# Resistance to Tyrosine Kinase Inhibitors in Hepatocellular Carcinoma (HCC): Clinical Implications and Potential Strategies to Overcome the Resistance

**DOI:** 10.3390/cancers16233944

**Published:** 2024-11-25

**Authors:** Ali Gawi Ermi, Devanand Sarkar

**Affiliations:** 1Department of Human and Molecular Genetics, Virginia Commonwealth University, Richmond, VA 23298, USA; ali.gawiermi@vcuhealth.org; 2Department of Human and Molecular Genetics, Massey Comprehensive Cancer Center, Virginia Commonwealth University, Richmond, VA 23298, USA

**Keywords:** hepatocellular carcinoma, sorafenib, tyrosine kinase, treatment resistance

## Abstract

Hepatocellular carcinoma (HCC), the primary liver cancer arising from hepatocytes, the predominant cell constituting the liver, accounts for ~90% of all liver cancers. HCC is characterized by activation of receptors on tumor cell surface which regulates tumor cell proliferation and spread to distant organs (metastasis). The class of drugs which predominantly inhibits these receptors are collectively known as tyrosine kinase inhibitors (TKIs). Among them sorafenib, lenvatinib, regorafenib, and cabozantinib have been approved either as the first or as the second line of treatment for advanced HCC. In general, ~30% of HCC patients respond to TKIs, and those that respond invariably develop resistance in ~6 months. This review paper addresses the potential molecular mechanisms by which resistance to TKIs develop and enumerates potential strategies to overcome them thereby providing prolonged survival benefit to HCC patients.

## 1. Introduction

### 1.1. Background on HCC: Epidemiology and Risk Factors

Hepatocellular carcinoma (HCC) accounts for the majority of primary liver cancers. Globally, liver cancer is the fourth leading cause of cancer-related deaths and ranks sixth in terms of incidence. According to the World Health Organization (WHO), it is anticipated that over 1 million patients will die from liver cancer by 2030 [1,2]. In the United States, the death rate from liver cancer increased by 43%, from 7.2 to 10.3 deaths per 100,000 people, between 2000 and 2016 [3,4]. With a 5-year survival rate of 18%, liver cancer is the second deadliest tumor after pancreatic cancer [5]. Most HCC develop in patients with pre-existing liver disease, primarily due to hepatitis B or C virus (HBV or HCV) infection or alcohol abuse [6]. Universal HBV vaccination and the widespread use of direct-acting antiviral agents against HCV are likely to alter the etiological landscape of HCC. However, the rise in metabolic dysfunction-associated fatty liver disease (MAFLD), a consequence of metabolic syndrome and obesity, increases the risk of liver cancer, and is expected to become a leading cause of liver cancer in Western countries [7]. Racial and ethnic differences significantly influence survival probabilities, with African-American and Hispanic individuals having increased susceptibility to HCC and are less likely to receive curative treatments compared to white individuals [8,9].

Future estimates indicate that the global burden of HCC will continue to be significant, with variations in trends depending on regional epidemiology and public health interventions. In the United States, for example, the incidence of HCC is expected to continue to increase until around 2030, driven by the rising prevalence of metabolic risk factors [10]. Conversely, regions with successful HBV vaccination and HCV treatment programs may see a decline in HCC incidence related to viral hepatitis [10]. HCC remains and will continue to remain in near future a major global health issue with substantial morbidity and mortality. Its prevalence and incidence are influenced by a complex interplay of viral, environmental, and lifestyle factors. Understanding these epidemiological trends is crucial for developing targeted prevention and treatment strategies to reduce the global burden of this deadly disease.

### 1.2. Current Treatment for HCC

Treatment for HCC varies based on disease stage, liver function, and patient health [11]. The choice of treatment is guided by several factors, including the stage of the cancer (early, intermediate, advanced), liver function (as assessed by the Child–Pugh score), performance status of the patient, and the presence of co-morbid conditions [12]. Treatment plans are often discussed in a multidisciplinary setting to tailor the best approach for each patient. For early and intermediate HCC, treatment options include (i) surgical resection where removal of the tumor is typically recommended for patients with a single tumor and good liver function; (ii) liver transplantation, which is suitable for patients with early-stage HCC and underlying liver disease, offering a chance to cure both the cancer and the liver disease; (iii) ablation techniques (radiofrequency, microwave, cryoablation) using heat to destroy cancer cells, which is effective for small tumors; and (iv) embolization procedures (chemoembolization, radioembolization) to deliver chemotherapy directly to the tumor, blocking its blood supply, or radiation therapy which involves injecting radioactive beads into the liver’s blood vessels to target the tumor [13]. In general, ~50% of HCC patients are diagnosed at advanced stage, which is managed by systemic therapies including targeted drugs and immunotherapy [13]. As a first line of treatment, systemic and targeted therapies predominantly include tyrosine kinase inhibitors (TKIs) such as sorafenib (Nexavar), an oral medication that inhibits multiple tyrosine kinases, lenvatinib (Lenvima), an alternative to sorafenib, and regorafenib (Stivarga) and cabozantinib (Cabometyx), used as second line of treatment for patients who have progressed on sorafenib [13,14,15,16,17,18,19]. Ramucirumab (Cyramza), a monoclonal antibody for vascular endothelial growth factor receptor 2 (VEGFR2), is recommended for patients with high alpha-fetoprotein levels [20]. However, a combination immunotherapy consisting of anti-programmed cell death ligand-1 (anti-PD-L1) (atezolizumab/Tecentriq) and anti-vascular endothelial growth factor (anti-VEGF) (bevacizumab/Avastin) antibodies demonstrated better overall and progression-free survival outcomes over sorafenib in treatment-naïve advanced HCC patients in the IMbrave150 trial, and is currently being recommended as the first line of treatment for advanced HCC [21]. However, the atezolizumab/bevacizumab combination has not been tested head-to-head with other TKIs, such as lenvatinib, and HCC patients must present with preserved liver function, properly controlled esophageal variceal bleeding, and should not have underlying vascular disorders, arterial hypertension, severe autoimmune disorders, and prior transplantation to qualify for atezolizumab/bevacizumab treatment [22]. As such, TKIs are still the first line of treatment for many advanced HCC patients, and in many countries in the world they are still extensively used because of lack of accessibility to immunotherapy [23,24]. As a single agent, an anti-programmed cell death-1 (anti-PD-1) antibody such as Nivolumab (Opdivo) or Pembrolizumab (Keytruda) is also used for advanced cases [25,26]. In terminal cases of HCC, who are not eligible for curative treatments, supportive care focuses on relieving symptoms and improving the quality of life for patients [13].

## 2. Importance of TKIs in HCC Treatment

TKIs play a crucial role in the treatment of HCC, especially in advanced stages of the disease for patients who are not candidates for curative treatments like surgery or liver transplantation. They offer a systemic treatment option that can be administered orally, providing a convenient treatment modality.

In HCC, tumor cell proliferation, metabolism, angiogenesis, and metastasis are mainly governed by out-of-control information transmission in the cells of the tumor, notably from the cell surface to the interior. The transmission of information is relayed by growth factors and their receptors, which are receptor tyrosine kinases (RTKs), such as epidermal growth factor receptor (EGFR), platelet-derived growth factor receptor (PDGFR), vascular endothelial growth factor receptor (VEGFR), hepatocyte growth factor receptor (HGFR)/MET, and fibroblast growth factor receptor (FGFR) [27]. The activation of these receptors upon engagement of the corresponding growth factors (Figure 1) further triggers the cascade of intra-cellular PI3K/AKT/mTOR and RAS/RAF/MEK/ERK protein kinase signaling, stimulating the whole process (Figure 2) [27].

The importance of TKIs in HCC treatment includes targeting these pathways, thereby inhibiting tumor growth, angiogenesis, and metastasis [28]. This multi-target approach disrupts several pathways critical for cancer progression. Indeed, clinical trials have demonstrated that TKIs can extend overall survival in patients with advanced HCC. Sorafenib, which acts on receptors such as VEGFR1-3, PDGFRβ, KIT proto-oncogene, receptor tyrosine kinase (KIT), RET protooncogene (RET), and FLT3 (fms-related receptor tyrosine kinase 3) as well as RAF1 and BRAF, was the first approved systemic therapy for HCC to show a survival benefit (Figure 3) [29]. The phase III, double-blinded, randomized trial SHARP was performed in 2007 in which 602 HCC patients were treated with sorafenib and demonstrated a significant increase in median overall survival by 2.8 months compared to placebo [14]. The phase III ASIA-PACIFIC study with 226 HCC patients from the Asia–Pacific region recapitulated the benefits identified in the SHARP trial [15]. REFLECT, a randomized, open, phase III clinical trial of 1492 HCC patients revealed no statistically significant difference in overall survival between lenvatinib- and sorafenib-treated patients, but progression-free survival was better for lenvatinib treatment, resulting in approval of lenvatinib as a first-line therapy [16]. Lenvatinib structure is significantly different from sorafenib, lacking the fluorine atoms of sorafenib, and it targets VEGFR1-3, PDGFRα/β, FGFR1-4, KIT, and RET (Figure 3) [29]. The availability of additional TKIs, such as regorafenib and cabozantinib, offers second-line treatment options extending the therapeutic arsenal for managing advanced HCC. Regorafenib’s chemical structure is similar to that of sorafenib, except having an extra fluorine atom to provide a wider range of targets to inhibit (Figure 3) [30]. Cabozantinib’s chemical structure is significantly different from sorafenib and it is a dual blocker of hepatocyte growth factor receptor (HGFR)/MET and VEGFR2, as well as inhibiting RET, KIT, TIE2/TEK, and AXL (Figure 3) [31]. Newer TKIs often have improved side-effect profiles compared to traditional chemotherapy, making them more tolerable for patients. This can lead to better adherence to treatment and improved quality of life. TKIs are also being evaluated in combination with other therapies, mainly immunotherapy. These combinations have shown promising results, providing enhanced efficacy through synergistic mechanisms. As understanding of HCC biology advances, TKIs can be part of a more personalized treatment approach. Biomarker studies are ongoing to identify patients who are more likely to respond to specific TKIs, potentially optimizing treatment outcomes. Overall, TKIs have significantly advanced the treatment landscape of HCC, providing effective options for managing a disease that is often diagnosed at an advanced stage. Their ability to target multiple pathways and their integration into combination regimens continue to enhance their role in HCC therapy.

## 3. Overview of Molecular Mechanisms of TKI Resistance in HCC

TKIs emerged as the first family of drugs providing significant increases in overall survival of the patients. However, long-term benefit from this treatment is minimal due to treatment failure caused by the development of resistance. Indeed, resistance to TKIs in HCC is an inevitable and universal consequence of exposure to TKIs and poses a significant challenge to effective treatment [32]. Resistance to TKIs can be broadly categorized into primary resistance and acquired resistance. In primary resistance, HCC patients are inherently non-responsive to TKIs mainly because of the genetic makeup and cellular constituents of the tumor. Indeed, only ~30% of HCC patients benefit from sorafenib, thus highlighting the importance of primary resistance [14]. In acquired resistance, the patients initially respond to the treatment, but the TKIs become progressively less effective and ultimately ineffective with the passage of time. The 30% of patients who originally respond to sorafenib, develop acquired resistance within 6 months [33]. Understanding the mechanisms behind this resistance is crucial for developing strategies to overcome it. Molecules/pathways mediating acquired resistance are activated or inhibited upon prolonged treatment with TKIs. However, these molecules/pathways might be activated or inhibited intrinsically thereby contributing to primary resistance. These mechanisms underlying primary and acquired resistance to TKIs are highlighted here. Because sorafenib was the first TKI to be used clinically, there is a vast body of literature interrogating sorafenib resistance. The mechanisms described in this section are broadly applicable to sorafenib. Specific examples of resistance to lenvatinib, regorafenib, and cabozantinib are described as well. There are numerous mechanisms by which resistance to drugs develop. TKIs are used in cancers other than HCC and resistance mechanisms to TKIs have been described in other cancers. In this review, we discuss TKI resistance mechanisms that have been identified in HCC without extrapolating observations from other cancers or other drugs.

### 3.1. Intra-Tumoral Heterogeneity

Tumors are notoriously heterogeneous, especially because of genomic instability [34]. The cancer cells continue to acquire mutations, which can be as simple as a point mutation to as drastic as development of polyploidy. As the tumor develops and progresses, alterations in dynamics within tumor environment drives the intra-tumoral heterogeneity. The presence of a selective pressure, such as hypoxia or nutrition requirement, causes appearance or disappearance of cell clones, acquisition of new mutations, and adaptation to changes in signaling mechanisms and epigenetic regulations [35]. Because of the inherent heterogeneous nature, not all cells in a tumor respond equally to treatment, which is true for TKI treatment as well, thereby accounting for some cases of primary resistance.

### 3.2. Inter-Tumoral Heterogeneity: EGFR

Apart from intra-tumoral heterogeneity, there are differences among HCC patients that underly inter-tumoral heterogeneity, contributing to primary resistance to TKIs. One well-studied mechanism of primary resistance to sorafenib is the activation of EGFR, with subsequent activation of RAS/RAF/MEK/ERK pro-survival signaling pathways [36,37]. EGFR activation provides compensatory signaling mechanisms, allowing cancer cells to escape from targeted therapies such as TKIs. Immunohistochemistry in tissue microarray identified 52.6% of HCC patients to be EGFR^high^ and with worse survival [38]. HCC cell lines with activated EGFR showed resistance to sorafenib [36,37]. These in vitro observations formulated the basis of the randomized trial SEARCH (Sorafenib and Erlotinib, a Randomized Trial Protocol for the Treatment of Patients with Hepatocellular Carcinoma) in which 720 systemic treatment-naïve advanced HCC patients were treated with sorafenib plus either EGFR inhibitor erlotinib (n = 362) or placebo (n = 358) [39]. However, no difference was observed in overall survival between these two treatment groups in advanced HCC patients. A small trial using 17 patients tested the combination of sorafenib and MEK inhibitor trametinib in treatment-naïve HCC patients, which demonstrated safety and tolerability of the combination, but did not show a significant anti-cancer effect [40]. A kinome-centered CRISPR/Cas-9 genetic screen identified that EGFR inhibition provided synthetic lethality to lenvatintib in HCC patients [38]. Preclinical in vivo studies using patient-derived xenografts demonstrated therapeutic benefits of a combination of lenvanitib and EGFR inhibitor gefitinib, followed by a small clinical trial using 12 lenvatinib-unresponsive advanced HCC patients, who showed meaningful clinical responses with the lenvatinib and gefitinib combination [38]. A larger clinical trial needs to be carried out to confirm these initial findings. The reasons for the negative results of the SEARCH trial are not clear. Whether erlotinib and gefitinib show differential response in HCC patients, and whether EGFR-activation provides compensatory pathways that are more relevant to Lenvatinib, remains to be seen. Additionally, it might be necessary to stratify the patients based on their EGFR levels to obtain meaningful benefits from such a combination.

### 3.3. Inter-Tumoral Heterogeneity: Genetic Mutations in Target Kinases and Alterations in Signaling Pathways

Tumor cells can acquire genetic mutations that alter the target kinases, rendering TKIs less effective. Mutations in the tyrosine kinase domain can prevent TKIs from binding effectively, leading to resistance. Sorafnib is a multi-kinase inhibitor. It can block BRAF, a serine threonine kinase that is part of the RAS/RAF/MEK/ERK pathway, and also VEGFR and PDGFR, which can lead to the activation of PI3K/AKT/mTOR as well as MEK/ERK MAPK pathways. Specific genetic mutations or variations in these target molecules as well as their levels of expression are crucial in developing resistance to TKIs in HCC. Increased pERK and VEGFR2 staining correlated with shorter progression-free and overall survivals in 77 HCC patients treated with sorafenib [41]. Analysis of 187 HCC patients identified amplifications of chromosome 6p21, which harbors VEGFA gene, in 11% of patients, and similar VEGFA amplification was also observed in mouse HCC [42]. It was documented that VEGFA-amplified mouse HCC were more sensitive to sorafenib, and HCC patients with a VEGFA gain showed markedly improved survival upon sorafenib treatment compared to non-VEGFA-gain patients [42]. It was suggested that enhanced angiogenesis in these patients made them uniquely sensitive to anti-angiogenic treatment like sorafenib. A retrospective multi-center study, ALICE-1 (Angiogenesis Liver CancEr), was aimed at evaluating VEGF and VEGFR polymorphisms in determining sorafenib treatment outcome in HCC patients [43]. VEGF-A alleles C of rs25648, T of rs833061, C of rs699947, C of rs2010963, VEGF-C alleles T of rs4604006, G of rs664393, VEGFR-2 alleles C of rs2071559, and C of rs2305948 were identified as significant predictors of progression-free survival (PFS) and overall survival (OS) in a univariate analysis, suggesting that genotyping prior to treatment might help stratify patients to increase the likelihood of benefit and avoid unnecessary toxicity [43].

A biomarker companion study (BIOSTORM), using tissues from phase 3 STORM trial on HCC patients treated with sorafenib or placebo, revealed that only pERK and microvascular invasion served as independent prognostic factors for poor recurrence-free survival (RFS) in sorafenib-treated patients [44]. This study also identified a 146-gene signature in 30% patients who benefitted from sorafenib treatment in terms of RFS [44]. Next-generation sequencing (NGS) of 127 HCC patients, 87 of whom received sorafenib treatment, identified activating mutations in the PI3K/mTOR pathway as a determinant of shorter median PFS and OS in sorafenib treatment [45]. This study did not identify activating mutations in the WNT or MAPK pathway to affect therapeutic outcome in these patients [45]. Genome-wide CRISPR/Cas9 library screening identified loss of neurofibromin 1 (NF1) and dual specificity phosphatase 9 (DUSP9) as potential regulators of lenvatinib resistance in HCC cells [46]. Using knockdown/knockout approaches, it was discovered that loss of NF1 reactivates the PI3K/AKT and MAPK/ERK signaling pathways, and loss of DUSP9 activates MAPK/ERK signaling with subsequent degradation of FOXO3, ultimately resulting in lenvatinib resistance [46]. MEK inhibitor trametinib could overcome resistance to lenvatinib induced by NF1 and DUSP9 loss [46].

In a small study analyzing 10 HCC patients who responded to sorafenib treatment, amplification of chromosome 11q13, which harbors FGF3 and FGF4 genes, was identified in 3 patients [47]. Although these patients responded to sorafenib, they were characterized by poor differentiation and multiple lung metastases, suggesting that even though the disease course is advanced, the dependence of the HCC on sorafenib targets makes them vulnerable to sorafenib treatment [47]. In vitro, cells with FGF3/FGF4 and FGFR2 amplifications showed hypersensitivity to sorafenib [47]. A potential role of FGF19/FGFR4 signaling in TKI resistance has been attributed in which HCC patients with high FGF19 levels showed shorter PFS and OS upon sorafenib treatment [48]. However, no such correlation was observed in lenvatinib-treated patients [48].

Sorafenib-resistant HuH-7 and HepG2 cells showed increased production of the hepatocyte growth factor (HGF) and overexpression of its receptor HGFR/MET, especially its active phosphorylated form, leading to the activation of AKT and ERK [49]. Combinatorial treatment with AKT and MET inhibitors, MK2206 and capmatinib, respectively, synergistically inhibited sorafenib-resistant xenografts in vivo [49]. A role of HGF/MET in mediating lenvatinib resistance has also been described [50]. The efficacy of cabozantinib was tested using multiple oncogene-driven mouse models [51]. It was shown that cabozantinib could inhibit disease progression in MET/β-catenin- and AKT/MET-driven models but not in AKT/RAS and MYC models [51]. While cabozantinib inhibited MET and ERK activities, it did not affect the AKT/mTOR pathway, and a combination of cabozantinib and mTOR inhibitior MLN0128 potentiated tumor regression in the MET/β-catenin-driven HCC model, suggesting that this combination might have a potential to overcome primary cabozantinib resistance [51]. Integrin subunit beta 8 (ITGB8) was shown to be upregulated in lenvatinib-resistant HCC cells [52]. ITGB8 induced lenvatinib resistance through heat-shock protein 90 (HSP90)-mediated stabilization of AKT and activation of the AKT signaling pathway [52]. AKT inhibitor MK-2206 or HSP90 inhibitor 17-AAG restored sensitivity of the resistant cells to lenvatinib [52].

AXL is a receptor tyrosine kinase functioning as an oncogene. AXL was upregulated in sorafenib- and lenvatinib-resistant HCC cells and protected these cells from immunogenic cell death by inhibiting TNF-α and STING-type I interferon pathways [53]. Interestingly, these resistant cells also showed poor response to anti-PD-1 antibody, and a combination of AXL inhibitor BGB324 and anti-PD-1 antibody suppressed in vivo tumor growth more robustly than each agent alone [53]. A potential role of synergistic activity of ErbB receptor and AXL was also shown in mediating regorafenib resistance [54].

Analysis of RNA-seq data from The Cancer Genome Atlas (TCGA) and Gene Expression Omnibus (GEO) identified differential expression of 827 mRNAs in acquired sorafenib-resistant HCC [55]. Functional enrichment analysis identified that modulation of these mRNAs led to the activation of the MAPK, JAK-STAT, TGFβ, and cytochrome P450 drug metabolism pathways [55]. Among these mRNAs, CDK1 (cyclin-dependent kinase 1), CDKN1A (cyclin-dependent kinase inhibitor 1A/p21), and TAPBP (transporter associated with antigen processing binding protein) were predicted to serve as prognostic markers for sorafenib resistance upon survival analysis [55]. Transcription factor-mRNA network analysis predicted 18 transcription factors regulating the expression of these differentially expressed mRNAs, among which NFKB1 and MYC were predicted to be prognostic for sorafenib resistance [55]. This in silico study establishes the way for developing combination treatments, e.g., combining NF-κB inhibitors, which are already being used clinically, with TKIs. Recently, a new class of specific MYC inhibitors have been identified that are yet to be tested in HCC and might be a component of this combinatorial treatment [56].

### 3.4. Epigenetic Changes

Epigenetic modifications induce changes in expression levels of genes without altering the DNA sequence and include chromatin remodeling, histone alterations, DNA/RNA methylation, and non-coding RNA (ncRNA) expression, all contributing to HCC development and progression [57]. Indeed, in tumors there is a reversible drug-tolerant subpopulation, called drug-tolerant expanded persisters (DTEPs), having an altered state of epigenetic reprogramming [58,59]. Here, we describe epigenetic modifications other than ncRNAs mediating TKI resistance. The role of ncRNAs are discussed in a separate section.

N6-methyladenosine (m6A) is a key RNA modification process regulating RNA stability and translation ability [60]. The system of m6A includes the writer RNA methyltransferase METTL3 and METT14 complex, the eraser RNA demethylases FTO and ALKBH5, and the reader m6A binding proteins, YTHDF2, IGF2BP1/2/3, YTHDC1, and YTHDF1/3 (Figure 4). In HCC cancer stem cells (CSCs), methyltransferase 3, N6-adenosine-methyltransferase complex catalytic subunit (METTL3)-mediated m6A modification of Frizzled-10 (FZD10) mRNA resulted in FZD10 upregulation [61]. FZD10 is a cell surface WNT receptor that activates the WNT/β-catenin pathway. By activating this pathway, along with YAP1, FZD10 augmented the CSC function and contributed to lenvatinib resistance, and a positive feedback loop between FZD10 and METTL3 was unraveled [61]. Inhibiting FZD10 by an adeno-associated virus or employing a β-catenin inhibitor returned lenvatinib sensitivity to these cells [61]. m6A reader YTH N6-methyladenosine RNA binding protein F1 (YTHDF1) was shown to bind to m6A-modified NOTCH1 mRNA, increasing its stability and translation, and thus resulting in expression of NOTCH1-target genes that promoted liver CSCs [62]. YTHDF1 conferred resistance to sorafenib and lenvatinib, and lipid nanoparticles delivering YTHDF1 siRNA enhanced sensitivity to both drugs in vivo [62]. m6A modification of peroxisome proliferator-activated receptor gamma coactivator-1α (PPARGC1A) by METTL3 resulted in PPARGC1A downregulation in lenvatinib-resistant PLC/PRF/5 cells [63]. PPARGC1A downregulation led to an increase in bone morphogenetic protein and activin membrane-bound inhibitor (BAMBI) and a decrease in acyl-CoA synthetase long-chain family member 5 (ACSL5), a regulator of ferroptosis, suggesting that PPARGC1A facilitates lenvatinib-induced ferroptosis [63]. Interestingly, metformin inhibited METTL3 and restored PPARGCA expression [63]. However, an in vivo study testing whether metformin can overcome lenvatinib resistance remains to be performed.

An unbiased proteomic screening analysis of lenvatinib-sensitive parental and lenvatinib-resistant HCC cells unraveled marked upregulation of methyltransferase-like protein-1 (METTL1) and WD repeat domain 4 protein (WDR4), which are components of the tRNA N^7^-methylguanosine (m^7^G) methyltransferase complex [64]. In vivo studies established the role of METTL1/WDR4 in conferring lenvatinib resistance, and it was demonstrated that METTL1 augmented translation of genes in the EGFR pathway [64].

### 3.5. Drug Efflux, Uptake, and Metabolism

Cancer cells have the propensity to overexpress membrane drug transporters facilitating efflux of drugs, a common mechanism of chemoresistance. ATP binding cassette (ABC) transporters are a family of proteins that use ATP to export specific compounds or flip them from the inner leaflet to outer leaflet of the cell membrane [65]. Increased expression of ABC transporters can pump TKIs out of the HCC cells, reducing their intra-cellular concentrations and effectiveness. It has been shown that ATP binding cassette subfamily G member 2 (ABCG2), also known as breast cancer resistance protein (BCRP), mediates sorafenib efflux, and in vitro treatment with ABCG2 inhibitor gefitinib augmented sensitivity to sorafenib [66]. In a clinical correlation study, 47 advanced HCC patients were given a single oral administration of sorafenib; plasma sorafenib levels were determined 3 h later and correlated with polymorphisms in ABCG2 and ATP binding cassette subfamily B member 1 (ABCB1), also known as P-glycoprotein (pgp). A significant association between low plasma sorafenib levels and ABCB1 3435C>T, ABCG2 34G>A, and ABCG2 1143C>T polymorphisms in heterozygous patients was identified, suggesting that these polymorphisms might determine sensitivity to this drug [67]. A polymorphism in ATP binding cassette subfamily C member 2 (ABCC2), 1249G>A, was shown to promote efflux of sorafenib [68]. It was shown that sorafenib treatment resulted in the induction of ABCB1 via activation of pregnane X receptor (PXR), which might contribute to acquired resistance to sorafenib [69].

The human solute carrier (SLC) family of transporters are major mediators of drug uptake, including TKIs, inside the cells [70]. In HCC patients, two alternative spliced variants and three single nucleotide polymorphisms (SNPs) in SLC22A1 gene, encoding organic cation transporter-1 (OCT1), were identified [71]. It was shown that R61S fs*10 and C88A fs*16 variants encoded truncated SLC22A1 proteins and lost their ability to transport sorafenib, thereby losing sorafenib sensitivity [71]. Low OCT1 mRNA levels correlated with increased survival in sorafenib-treated patients in a study analyzing 60 HCC patients [72]. A retrospective analysis of 39 HCC patients, treated with sorafenib, revealed a correlation of decreased OCT1 level at the plasma membrane, rather than total OCT1 levels, with better outcome in sorafenib-treated patients [73]. Low levels of SLC46A3 were also shown to be a determinant of sorafenib resistance [74].

Cytochrome P450 family 3 subfamily A member 4 (CYP3A4) metabolizes more than half of all clinically used drugs, and increased expression and activity of CYP3A4 leads to decreased systemic levels of drugs, thus reducing drug efficacy. Sorafenib treatment leads to CYP3A4 induction in HCC throughout the course of treatment, resulting in a decline in systemic sorafenib levels, which might be a potential mechanism of acquired sorafenib resistance [75]. Similar decline in systemic sorafenib levels during the course of the treatment was also observed in 15 HCC patients [76]. Polymorphisms in CYP enzymes also contribute to sorafenib efficacy; e.g., the CYP3A5*3 genotype in the Chinese population exhibited minimal sorafenib metabolism, leading to hepatic and renal toxicity [77].

### 3.6. Cancer Stem Cells (CSCs)

CSCs or tumor-initiating cells are a subpopulation of tumor cells with self-renewal capabilities and are often more resistant to therapies including TKIs. HuH-7 xenografts in mice were treated with sorafenib, and sorafenib-sensitive and acquired sorafenib-resistant tumors were identified [78]. These sorafenib-resistant tumors showed significant enrichment of CSCs with concomitant activation of IGF and FGF signaling pathways [78]. In vivo, FGFR inhibitor brivanib significantly prolonged the survival of sorafenib-resistant tumors compared to sorafenib [78]. CSCs isolated from human HCC cells showed significant resistance to sorafenib, and sorafenib treatment resulted in further activation of the ERK and AKT signaling pathways in these cells [79]. RNA-sequencing analysis identified enrichment of the cholesterol biosynthesis pathway in sorafenib- and lenvatinib-resistant patient-derived xenografts (PDXs) as well as in CD133+ CSCs in these resistant tumors [80]. SREBP2, a key regulator of cholesterol biosynthesis, was upregulated in the CSCs [80]. In mechanistic studies, it was identified that sorafenib- and lenvatinib-resistant cells presented with increased caspase-3 activity, which cleaved SREBP2 from ER membrane to drive cholesterol synthesis that contributed to TKI resistance by activating the sonic hedgehog signaling pathway [80]. Indeed, the cholesterol-synthesis inhibitor simvastatin in combination with sorafenib could suppress tumor growth of sorafenib-resistant cells [80].

The WNT/β-catenin pathway plays a key role in regulating differentiation and proliferation of stem cells. Leucine-rich repeat-containing G-protein-coupled receptor 5 (LGR5)-positive CSCs, promoting HCC and showing marked sorafenib-resistance, expressed high levels of lysine-specific demethylase-1 (LSD1/KDM1A), which inhibited expression of several β-catenin suppressors, such as APC and Prickle 1 by regulating mono- and di-methylation of histone H3 lysine-4 as the promoters of these genes, and thereby augmented β-catenin activation [81]. KDM1A inhibitors, such as pargyline and GSK2879552, decreased stem-like properties of sorafenib-resistant HCC cells, inhibited β-catenin activity, and resensitized these cells to sorafenib in vivo [82]. CD13 is a marker for CSCs and it was shown that CD13 interacted with histone deacetylase 5 (HDAC5), resulting in HDAC5-mediated deacetylation of LSD1 and protein stabilization with consequent demethylation of p65 subunit of NF-κB, leading to increased p65 stability [83]. Activation of this molecular pathway contributed to sorafenib resistance, and CD13 inhibitor ubenimex, in combination with sorafenib, abrogated sorafenib resistance in HCC PDX models [83].

In sorafenib-resistant HCC PDXs, EPH receptor B2 (EPHB2) was found to be highly expressed, especially in the CSC population [84]. It was shown that EPHB2 activates the SRC/AKT/GSK3β/β-catenin signaling pathway and EPHB2 itself is upregulated by TCF1, thus establishing a positive feedback loop with the WNT/β-catenin pathway [84]. Adeno-associated virus 8 (AAV8)-mediated delivery of EPHB2 shRNA sensitized HCC tumors to sorafenib in an NRAS/AKT-driven immunocompetent model of mouse HCC [84]. A kinome profiling screen in lenvatinib-resistant HCC cells identified cyclin-dependent kinase 6 (CDK6) upregulation via the ERK/YAP1 signaling pathway [85]. It was shown that CDK6 augmented liver CSCs and activated WNT/β-catenin signaling by a non-canonical pathway in which it bound to and modulated GSK3β activity [85]. CDK6 inhibitor palbociclib or proteolysis targeting chimeras (PROTAC) for CDK6 could synergize with lenvatinib in lenvatinib-resistant tumors [85].

### 3.7. Epithelial–Mesenchymal Transition (EMT)

EMT is a process where epithelial cancer cells acquire mesenchymal features, which are associated with increased motility and invasiveness in many cancer cells, including HCC. EMT can confer resistance to TKIs by altering cellular signaling and enhancing survival pathways. E3 ubiquitin ligase TRIM15 was found to be upregulated in sorafenib- and regorafenib-resistant HuH-7 and Hep3B cells via activation of AKT and FOXO1 [86]. It was shown that TRIM15 interacted with LIM and SH3 protein 1 (LASP1) resulting in its K-63-linked polyubiquitination and nuclear translocation, leading to upregulation of Snail, induction of EMT, and TKI resistance [86]. Sorafenib-resistant HuH-7 and Hep3B cells showed a EMT phenotype, and the transcription factor E26 transformation-specific-1 (ETS1) was shown to be upregulated in these cells, regulating EMT-related genes [87]. Additionally, ETS1 provided protection from mitochondrial damage and reactive oxygen species (ROS) production by inducing the expression of glutathione peroxidase 2 (GPX2) [87]. This finding creates the scientific premise to develop PX2-specific inhibitors, which might be used to overcome resistance to TKIs.

TFGβ is a key regulator of EMT and it was shown that TGFβ treatment of human HCC cells PLC/PRF/5 resulted in upregulation of multiple RTKS, such as IGF1R, EGFR, PDGFBR, and FGFR1, resulting in sorafenib resistance [88]. A TGFβR1 inhibitor LY2157299 augmented sorafenib-induced apoptosis in vitro [88].

### 3.8. Hepatic Cells Metabolic Reprogramming and Hypoxia

A hallmark of cancer cells is increased anerobic glycolysis and decreased oxidative phosphorylation, the Warburg effect [89]. To identify regulators of sorafenib resistance, a CRISPR/Cas9 knockout library screening was performed in MHCC97L cells, which identified upregulation of phosphoglycerate dehydrogenase (PHGDH), the first enzyme in the serine synthesis pathway (SSP) [90]. It was shown that sorafenib treatment induced PHGDH and activated SSP, and a PHGDH inhibitor NCT-503 synergized with sorafenib to inhibit in vivo growth of HCC tumors [90]. Additionally, PHGDH was shown to mediate resistance to regorafenib and lenvatinib [90]. SSP in collaboration with the folate cycle generates many important metabolites, including purines; other amino acids like glycine, methione, and cysteine; NADPH; α-ketoglutarate; and sphingosine [91]. Interestingly, serine per se could not confer TKI resistance to HCC cells, suggesting that additional metabolites generated through SSP might mediate the resistance [90]. In HuH-7 cells, a CRISPR/Cas9 screen to identify genes mediating regorafenib resistance identified hexokinase 1 (HK1), which catalyzes the first step in glycolysis [92]. It was shown that HK1 also mediates sorafenib resistance, and sorafenib treatment of human HCC patient-derived xenografts (PDXs) induced HK1 and HK1 levels correlated with tumor growth [92]. The glycolysis inhibitor lonidamine resensitized regorafenib-resistant cells to regorafenib [92]. Enhanced glycolysis was also observed in lenvatinib-resistant HuH-7 cells [93]. In these cells, mitophagy mediated by increased activation of BCL2 interacting protein 3 (BNIP3) shifted energy production from mitochondrial oxidative phosphorylation to glycolysis via AMP-activated protein kinase (AMPK)-enolase 2 (ENO2) signaling pathway [93]. In vivo, the BNIP3 inhibitor olomoucine cooperated with lenvatinib to inhibit growth of lenvatinib-resistant HCC cells [93]. Metabolomics analysis of regorafenib-resistant HuH-7 and Hep3B cells revealed that the pentose phosphate pathway (PPP) is the most relevant metabolic pathway in these cells with increased glucose-6-phosphate dehydrogenase (G6PD) activity and NADPH/NADP^+^ ratio [94]. It was shown that G6PD activation led to activation of the PI3K/AKT pathway and a G6PD inhibitor, 6-aminonicotinamide, potentiated regorafenib-mediated killing in the resistant cells [94]. Histone deacetylase 11 (HDAC11) is overexpressed in HCC. Using an HDAC11 conditional knockout mouse, it was shown that loss of HDAC11 promoted histone acetylation of the promoter of serine/threonine kinase 11 (STK11/LKB1) gene, resulting in its increased transcription and leading to activation of the AMPK signaling pathway and inhibition of glycolysis [95]. Low levels of HDAC11 correlated with increased overall survival in sorafenib-treated HCC patients and overexpression of HDAC11 conferred resistance to sorafenib [95]. In general, regulation of glycolysis levels might be one mechanism by which response to TKIs can be improved. The metabolic changes contributing to TKI resistance are highlighted in Figure 5.

The TME of HCC is highly hypoxic. In normal human liver, the median partial pressure of oxygen (pO_2_) is 30 mm Hg while that in HCC is ~6 mm Hg [96]. Lack of oxygen availability due to inadequate blood supply resulting from poor vasculature in the tumor and increased oxygen consumption from metabolically active HCC cells lead to intra-tumoral hypoxia. Tumor regions proximal to a blood vessel are oxygenated, and oxygen level gradually decreases away from the blood vessels forming hypoxic regions. The tumor core is usually extremely hypoxic and is characterized by necrosis. TKIs further contribute to tumor hypoxia by inhibiting pro-angiogenic RTKs, such as VEGFR, PDGFR, and FGFR. In hypoxic conditions, lack of the ultimate electron acceptor oxygen causes inhibition in oxidative phosphorylation with resultant increase in glycolysis. HIF-1α is induced by hypoxia and drives the expression of many genes that regulate these metabolic processes. Indeed, HIF-1α-mediated glycolysis has been shown to contribute to sorafenib resistance in HCC cell lines [97]. The glucose transporters, GLUT1 and GLUT3, and hexokinase 2 (HK2), an enzyme catalyzing the first step of glycolysis, are HIF-1α-target genes and are upregulated in sorafenib-resistant HCC cells [97,98,99,100]. Inhibiting glycolysis by 2-deoxyglucose (2-DG) significantly augmented sorafenib sensitivity in sorafenib-resistant HCC cells [101]. Similarly, inhibition of HK2 by 3-bromopyruvate (3-BP) augmented efficacy of sorafenib on HCC cells under a hypoxic condition [100].

### 3.9. Autophagy and Inhibition of Ferroptosis

Autophagy has a dual effect. On one hand, under stressful and nutrition-deprived conditions, autophagy-mediated degradation of internal organelles provides basic building blocks of metabolism that support cell survival (protective autophagy) [102]. On the other hand, excessive autophagy, induced by drugs, can lead to cytotoxic autophagy, resulting in cell death [102]. Whether TKI-induced autophagy is cytoprotective or cytotoxic depends upon context, and studies have shown involvement of both in TKI resistance. Sorafenib treatment induced both autophagy and apoptosis in human HCC cells [103]. Sorafenib first induced endoplasmic reticulum (ER) stress, which led to protective autophagy and the autophagy inhibitor chloroquine synergized with sorafenib in inhibiting HCC tumors in vivo [103]. Sorafenib-resistant HepG2 and HuH-7 cells showed increased activation of AKT and increased autophagy [104]. Consequently, autophagy inhibition increased sensitivity to sorafenib in sorafenib-resistant cells, while autophagy activation by rapamycin exerted the opposite effect [104]. AKT inhibition by GDC0068 synergized with sorafenib in inhibiting growth of sorafenib-resistant tumors by switching cytoprotective autophagy to cytotoxic autophagy [104]. A whole genome CRISPR/Cas9 knockout library screen identified overexpression of lysosomal protein transmembrane 5 (LAPTM5) contributing to lenvatinib resistance in HuH-7 cells [105]. LAPTM5 increased intrinsic macroautophagy and autophagy inhibitor hydroxychloroquine and lenvatinib synergistically inhibited in vivo tumor growth [105]. Analysis of lenvatinib-treated HCC patient samples showed that LAPTM5 levels correlated with lenvatinib sensitivity [105]. Neurotransmitter signaling, e.g., activation of the β-adrenergic receptor, can promote cancer initiation and progression. Activation of the β2-adrenergic receptor (ADRB2) by adrenaline resulted in AKT activation with subsequent disruption of the Beclin1/VPS34/Atg14 complex, thereby inhibiting autophagy [106]. As a consequence, autophagic degradation of HIF1α was abrogated, resulting in increased stability of HIF1α and increased glucose metabolism [106]. ADRB2 signaling inhibited sorafenib-induced autophagy, thereby contributing to sorafenib resistance, and ADRB2 inhibitor ICI118551 promoted sorafenib-induced autophagy and inhibition of colony formation by HCC cells in vitro [106]. Annexin A3 (ANXA3), a member of the annexin family of Ca^2+^-dependent phospholipid-binding proteins, was found to promote aggressive HCC with stem cell-like properties as well as resistance to chemotherapy [107]. Analysis of sorafenib-resistant HepG2 and HuH-7 cells identified increased expression of ANXA3, which inhibited PKCδ/p38 MAPK-mediated apoptosis and increased cytoprotective autophagy [108]. ANXA3 levels correlated with worse overall survival in sorafenib-treated HCC patients, and combination of anti-ANXA3 monoclonal antibody and sorafenib or regorafenib inhibited HCC tumor growth in vivo with significant increase in survival [108].

Ferroptosis is a type of programmed cell death which is dependent on iron and is characterized by the accumulation of lipid peroxides and iron in the cell [109]. Sorafenib is known to induce ferroptosis in HCC cells [110]. It was shown that sorafenib treatment induced metallothionein 1G (MT1G), a cysteine-rich protein functioning in heavy metal detoxification and as an antioxidant, via activation of the transcription factors nuclear factor erythroid 2-related factor 2 (NRF2) and HIF1α [111]. MT1G inhibition, genetic or pharmacological, augmented sorafenib activity in xenograft assays [111]. It was shown that MT1G inhibited sorafenib-induced ferroptosis, thereby contributing to sorafenib resistance [111]. However, it was not checked whether MT1G level is persistently induced in sorafenib-resistant HCC cells. The leukemia inhibitory factor receptor (LIFR) is a tumor suppressor gene which is downregulated in HCC. Loss of LIFR activates NF-κB signaling, resulting in upregulation of the iron-sequestering cytokine LCN2, which depletes iron and induces resistance to sorafenib-induced apoptosis [112]. It was shown that LCN2-neutralizing antibody enhanced sorafenib-induced apoptosis in HCC PDXs with low LIFR and high LCN2 levels [112].

### 3.10. Non-Coding RNAs

Several miRNAs and long non-coding RNAs (lncRNAs) have been documented to regulate sensitivity or resistance to sorafenib. The PI3K/AKT and MAPK pathways are frequently activated in HCC and are known to modulate sorafenib resistance and the miRNAs that regulate these pathways, thereby apoptosis and autophagy, and also modulate sorafenib resistance [113,114,115,116]. KRAS, which lies upstream from the PI3K/AKT and MEK/ERK pathways, is increased in HCC patients and KRAS upregulation is associated with sorafenib resistance [114]. miR-622 directly targets KRAS and its downregulation is associated with sorafenib resistance [114]. As a corollary, sorafenib resistance could be overcome by a KRAS inhibitor or miR-622 mimic [114]. Ras association domain family member 1 (RASSF1), a negative regulator of MAPK signaling, is targeted by miR-181a [115]. In HepG2 and Hep3B cells, miR-181a was shown to confer sorafenib resistance because of its ability to target RASSF1 and thereby activating MAPK signaling [115]. MAPK4K3 is targeted by miR-1991-5p and let-7C, both of which are downregulated in HCC cells and a combination of these two miRNAs was shown to augment sorafenib sensitivity in vitro [116]. The PI3K/AKT pathway is negatively regulated by the phosphatase and tensin homolog (PTEN), which is targeted by multiple miRNAs, such as miR-21 [117]. Comparison between sorafenib-resistant and sorafenib-sensitive clones of HuH-7 cells identified miR-21 upregulation in the resistant cells and anti-miR-21 augmented sorafenib-induced autophagy in both in vitro studies and in vivo xenograft assays [117]. Upregulation of the miR-216a/217 cluster is observed in HCC tissue samples [118]. This cluster targets SMAD family member 7 (SMAD7) and PTEN thus activating TGF-β and PI3K/AKT signaling, respectively, and its overexpression contributes to sorafenib resistance [118]. Several other miRNAs, such as miR-222, miR-93, and miR-494, target PTEN and their overexpression contributes to sorafenib resistance [119,120,121]. However, more in-depth studies are needed to unravel their role in acquired sorafenib resistance. On the other hand, miR-7 targets the receptor tyrosine kinase TYRO3, which activates the PI3K/AKT pathway, and miR-7 is downregulated in sorafenib-resistant HuH-7 cells [122]. In these cells, overexpression of miR-7 restored sensitivity to sorafenib [122].

miRNAs targeting apoptosis- and autophagy-regulating genes have been implicated in sorafenib resistance. miR-221 was identified to be upregulated in sorafenib-resistant HCC nodules in a rat model and mediated resistance to sorafenib-induced apoptosis by targeting caspase-3 [123]. Several miRNAs target anti-apoptotic proteins, and downregulation of these miRNAs was shown to be associated with sorafenib resistance. These miRNA/targets include miR-193b/Mcl-1, miR-34a/Bcl-2, and let-7 family/Bcl-xL [124]. For miR-193b, the downregulation was induced by stable integration of the HBV genome in HepG2 cells [124,125,126]. miR-142-3p targets autophagy-related 5 (ATG5) and autophagy-related 16-like (ATG16L1), and switched sorafenib-induced protective autophagy to sorafenib-induced apoptosis in in vivo xenograft assays [127].

Using a miRNA microarray in sorafenib-resistant HuH-7 and PLC/PRF/5 cells, downregulation of miR-122 was identified with consequent upregulation of its target IGF-1R and activation of IGF signaling [128]. In a separate study, miR-122 downregulation caused sorafenib resistance by upregulating its target solute carrier family 7 member 1 (SLC7A1), an arginine transporter, resulting in increased levels of nitric oxide [129]. Additional miRNAs/targets conferring sorafenib sensitivity include miR-486/Rho-interacting serine/threonine kinase (CITRON) and Claudin 10 (CLDN10), and miR-367-3p/MDM2 and miR-338-3p/Hypoxia-inducible factor-1 (HIF-1α), although convincing evidence of their roles in sorafenib resistance remains to be determined [130,131,132]. In sorafenib-resistant HepG2 cells, miR-744 was downregulated with consequent upregulation of paired box 2 (PAX2) [133]. However, the exact mechanism by which PAX2 is involved in sorafenib resistance was not elucidated. Interestingly, miR-744 is an exosomal miRNA that was decreased in HCC patient sera as well as in exosomes derived from sorafenib-resistant cells [133]. The role of this secreted miRNA in sorafenib resistance needs further exploration. Some of the important miRNAs that mediate sorafenib resistance are highlighted in Table 1.

lncRNAs, longer than 200 nucleotides, have been shown to contribute to sorafenib resistance by sponging miRNAs and increasing the levels of their corresponding targets. Metastasis-associated lung adenocarcinoma transcription 1 (MALAT1) sponges miR-140-5p and increases its target Aurora kinase A [134]. This mechanism was attributed to MALAT1-mediated sorafenib resistance in HepG2 and SMMC-7721 cells [134]. In vivo studies unraveled that MALAT1 knockdown restored sorafenib sensitivity [134]. THOR (testis-associated highly conserved oncogenic long non-coding RNA) stabilizes β-catenin and facilitates expansion of CSCs [135]. In vitro, knockdown of THOR augmented sorafenib sensitivity [135]. Nuclear paraspeckle assembly transcript 1 (NEAT1) activates the c-Met/AKT pathway by sponging miR-335, and in HepG2 xenografts NEAT1 knockdown enhanced sensitivity to sorafenib [136]. In HCC cells, HBV protein HBx upregulates translation regulatory lncRNA1 (TRERNA1), which upregulates NRAS and activates the RAF/MEK/ERK pathway by sponging miR-22-3p, thereby conferring sorafenib resistance [137]. Sorafenib resistance is induced by the oncogenic transcription factor FOXM1, which augments transcription of LINC-ROR (long intergenic non-protein coding RNA, regulator of reprogramming) [138]. Interestingly, LINC-ROR was shown to be in a positive feedback loop in which it increased FOXM1 by sponging miR-876-5p [138]. Single-cell RNA-sequencing performed in sorafenib-resistant HuH-7 cells identified lncRNA ZFAS1 (ZNFX1 antisense RNA 1) as the most abundant transcript [139]. In HCC patients, ZFAS1 levels correlated with levels of stemness and EMT-regulating genes, known to be associated with sorafenib resistance in HuH-7 and Hep3B cells, and siRNA-mediated ZFAS1 knockdown made sorafenib-resistant cells sensitive [139]. It remains to be studied whether the effect of ZFAS1 on stemness and EMT genes is mediated by sponging of any specific miRNA or if there are additional mechanisms. In HepG2 and MzChA1 cells, linc-VLDLR is released in extracellular vesicles (EVs), and this release was augmented by sorafenib treatment [140]. Interestingly, treatment of HCC cells with these EVs protected from sorafenib-induced death [140]. Mechanistically, linc-VLDLR knockdown resulted in downregulation of ABCG2 and increased sorafenib sensitivity, and ABCG2 overexpression rescued this effect [140]. The molecular link between linc-VLDLR and ABCG2 was not identified in this study. lncRNA BBOX1-AS1 sponged miR-361-3p thereby increasing PHF8, a histone lysine demethylase contributing to HCC progression and resistance to sorafenib, the underlying mechanism of which remained to be explored [141]. Sorafenib induces the iron-dependent programmed cell death ferroptosis. lncRNA URB1-antisense RNA 1 (URB1-AS1) was induced in sorafenib-resistant cells by HIF-1α [142]. URB1-AS1 interacted with ferritin, resulting in its phase separation and thereby regulating ferroptosis; in vivo studies showed that knocking down URB1-AS1 restored sorafenib sensitivity [142].

In sorafenib-resistant HepG2, SK-HEP1, HuH-7, and LM3 cells, circular RNA circRNA-SORE was upregulated, which contributed to sorafenib-resistance by sponging miR-103a-2-5p and miR-660-3p and activating the Wnt/β-catenin signaling pathway [143]. The increase in circRNA-SORE in sorafenib-resistant cells was mediated by increased m6A modification, resulting in increased RNA stability [143]. In vivo, a lentivirus delivering shRNA for circRNA-SORE augmented sorafenib sensitivity of sorafenib-resistant xenografts [143]. circRNA-SORE was also shown to bind to the oncoprotein YBX1, preventing its degradation, and circRNA-SORE could be transported via exosome to neighboring cells to induce sorafenib resistance [144]. circFOXM1 was identified to be upregulated in sorafenib-resistant HepG2 and HuH-7 cells, and it was shown that circFOXM1 mediates sorafenib resistance by sponging miR-1324, resulting in upregulation of its target methyl-CpG binding protein 2 (MECP2), a known oncogene [145]. Circular RNA cirDCAF8 is upregulated in human HCC samples and it promoted tumorigenesis by sponging miR-217 with resultant increase in its target nucleosome assembly protein 1 like 1 (NAP1L1), which serves as an oncogene [146]. It was demonstrated that cirDCAF8 was released from regorafenib-resistant HepG2 and Hep3B cells in exosome, and exosomal cirDCAF8 could be transferred from regorafenib-resistant cells to regorafenib-sensitive cells, rendering the latter regorafenib-resistant [146]. The mechanism by which cirDCAF8 confers regorafenib resistance remained to be explored. lncRNAs and circRNAs contributing to TKI resistance are summarized in Table 2.

## 4. Tumor Microenvironment and TKI Resistance

The tumor microenvironment (TME) plays a crucial role in the development of resistance to TKIs in HCC. The TME consists of various cellular and non-cellular components that interact with tumor cells, influencing their behavior and response to treatment. Key aspects of the TME contributing to TKI resistance include the following factors:

### 4.1. Cancer-Associated Fibroblasts (CAFs)

CAFs directly interact with cancer cells and secrete a variety of growth factors, chemokines, and cytokines that facilitate tumor growth, metastasis, suppression of anti-tumor immune response, and drug resistance. Using a 3D organoid co-culture system of primary HCC and CAFs from both mouse and human, it was shown that organoids co-cultured with CAFs demonstrated resistance to sorafenib and regorafenib, although the underlying molecular mechanism was not explored [147]. In a separate study, it was shown that CAFs but not normal fibroblasts released the chemokine CXCL12, which upregulated folate receptor alpha (FOLR1) in HCC cells and augmented autophagy, thus contributing to sorafenib resistance [148].

### 4.2. Immune Cells

Tumor-associated macrophages (TAMs) can be either pro-inflammatory and anti-tumorigenic M1 type or anti-inflammatory and pro-tumorigenic M2 type. It was shown that more M2 TAMs accumulate in sorafenib-resistant tumors compared to sorafenib-sensitive tumors [149]. These M2 TAMs secrete HGF, thereby activating the HGF/MET/ERK and PI3K/AKT pathways in the tumor cells and thus facilitating sorafenib resistance [149]. Additionally, HGF functioned as a chemoattractant for other macrophages, thus exacerbating the process [149]. However, the mechanism of M2 polarization of TAMs in sorafenib-resistant TME remains to be determined. CCL2+ or CCL17+ tumor-associated neutrophils (TANs) were observed in the stroma of HCC, and conditioned media from TANs or recombinant CCL2 or CCL17 increased migration of macrophages and regulator T cells (Tregs) [150]. CXCL5, released from HCC cells, promoted migration of TANs under hypoxic condition [150]. Sorafenib treatment increased TANs in the tumors and combination of sorafenib with TAN depletion by anti-Ly6G antibody inhibited tumor growth more than each agent alone, which was accompanied by marked inhibition of angiogenesis [150]. Tregs create an immunosuppressive environment and it was shown that CCR4+ Tregs were associated with HBV-positive HCC, contributing to sorafenib resistance [151]. A CCR4 antagonist N-CCR4-Fc interfered with intra-tumoral Treg accumulation, facilitated overcoming sorafenib resistance, and also enhanced efficacy of anti-PD-1 immunotherapy [151]. The mechanism by which CCR4+ Tregs contribute to sorafenib resistance remains to be studied.

Some of the important mechanisms of TKI resistance and strategies to overcome them are listed in Figure 6.

## 5. Strategies to Overcome TKI Resistance in HCC

Many approaches have been developed to overcome resistance to TKIs in HCC treatment, which include development of new therapies, combination treatments, and personalized medicine. However, there is a major obstacle in translating findings from laboratory and pre-clinical research to actual clinical trials. In many instances, molecules and/or pathways that are activated in TKI-resistant HCC cells are identified and then combinations of small molecule inhibitors targeting those molecules/pathways with TKIs are tested in in vivo xenograft studies in mice. The combinations restore TKI sensitivity to the resistant cells, resulting in synergistic suppression of tumor growth. Most of these studies are performed in subcutaneous or orthotopic xenograft experiments in immunocompromised mice using either established cell lines or HCC PDXs. A few studies have also been performed using an orthotopic allograft model in immunocompetent syngeneic mice. However, apart from the tumors growing in the liver, the architecture and function of the rest of the liver are preserved, which is distinct from most HCC patients in which the tumor develops on a fibrotic/cirrhotic background upon long-standing chronic inflammation induced by viral or alcoholic hepatitis or MASH. TKIs are inherently toxic with significant adverse effects, which is exacerbated in HCC patients in which a liver with compromised function fails to properly metabolize the drugs. As such, in a patient who has advanced HCC and who has already developed resistance to TKI, exposure to a combinatorial treatment may exert significant side effects resulting in discontinuation of treatment. Findings from pre-clinical xenograft studies should be considered carefully and small-scale pilot studies on the toxicity profile must be performed in relevant HCC patients before carrying out efficacy studies. Nevertheless, research into potential strategies to overcome TKI resistance has made significant in-roads.

### 5.1. Combination Therapies

Many combination strategies using TKIs with immunotherapy, other targeted therapies, and chemotherapy have been evaluated in in vitro studies or in pre-clinical xenograft models. Here, we highlight those approaches that have either moved to the arena of clinical trials or have shown promise in in-depth pre-clinical studies. Other clinical trials are not discussed here.

#### 5.1.1. TKIs and Immune Checkpoint Inhibitors (ICIs)

Programmed cell death protein 1 (PD-1) and cytotoxic T-lymphocyte-associated protein-4 (CTLA-4) are immune checkpoints that abrogate anti-tumor immune response [152]. PD-1 predominantly induces T-cell exhaustion in the TME, while CTLA-4 inhibits activated and regulatory T cells in the lymphoid organs [152,153]. Anti-PD-1 antibodies, such as nivolumab and pembrolizumab, serve as ICIs and are approved as second line treatment for HCC patients previously treated with sorafenib [25,26]. CheckMate 040 is an open-label, multi-cohort, phase I/II study evaluating nivolumab alone or in combination with other agents in advanced HCC patients who were previously treated with sorafenib or are intolerant to sorafenib [25]. Patients were recruited from 31 centers in 10 countries from Asia, Europe, and North America. As a monotherapy, nivolumab showed manageable toxicity, 14% overall response rate (ORR), at least 12 months of response duration in 59% patients and 15.1 months median survival in sorafenib-treated advanced HCC patients [25]. In another cohort of 148 such patients, combination of nivolumab with an anti-CTLA-4 antibody, ipilimumab, demonstrated manageable safety, 32% ORR, and 22.8 months median overall survival, resulting in accelerated approval of this combinatorial treatment in the US [154]. In Cohort 6 of the CheckMate 040 study, nivolumab and cabozantinib combination was compared with nivolumab+cabozantinib+ipilimumab combination in 71 advanced HCC patients who were either treatment-naïve, sorafenib-intolerant, or progressed on sorafenib [155]. Both in sorafenib-naïve and sorafenib-pretreated patients, either combination showed manageable safety, with ORR of 17% and 29%, and median OS of 20.2 and 22.1 months in the doublet and triplet combinations, respectively [155]. It should be noted that many trials have been performed or are ongoing to evaluate TKI and immunotherapy combinations in treatment-naïve advanced HCC patients to develop a combinatorial strategy as the first line of treatment [156]. Indeed, the currently approved first-line treatment for advanced HCC include the combination of anti-PD-L1 antibody atezolizumab and anti-VEGF antibody bevacizumab, and the IMbrave150 clinical trial, which demonstrated its superiority over sorafenib in terms of OS and PFS, was performed in treatment-naïve patients [21]. It is necessary to check the efficacy of immunotherapy combinations, including atezolizumab and bevacizumab combination, in TKI-resistant patients, to develop strategies to overcome resistance.

#### 5.1.2. TKIs and Other Targeted Therapies

EGFR activation has been implicated to confer primary resistance to TKIs [36,37]. In the SEARCH trial, performed in treatment-naïve advanced HCC patients, however, sorafenib and EGFR inhibitor erlotinib combination did not fare better against placebo, a meaningful clinical response was observed with lenvatinib and EGFR inhibitor gefitinib combination in a small study in 12 lenvatinib-unresponsive patients [38,39]. CDK6 has been shown to mediate lenvatinib resistance [85]. HCC was induced in C57BL/6 mice by hydrodynamic injection of MYC and sgRNA for p53, and the mice were treated with lenvatinib until the tumors developed resistance to lenvatinib and continued to grow [85]. In this model, combination of CDK6 inhibitor palbociclib with lenvatinib showed the maximal suppression of tumor growth, warranting evaluation of this combination in clinical trials with TKI-resistant advanced HCC patients [85].

### 5.2. Targeting Hypoxia

Hypoxic microenvironments favor TKI resistance and using agents that target HIFs can reduce the adaptive response of tumor cells to hypoxic conditions, potentially enhancing the efficacy of TKIs. EF24, a molecule structurally similar to curcumin, inhibited HIF1α by sequestering it in the cytoplasm and inducing its degradation by upregulating the Von Hippel–Lindau (VHL) protein [98]. EF24 synergized with sorafenib in the orthotopic HuH-7 xenograft model [98]. miR-338-3p targets HIF1α and overexpression of miR-338-3p sensitized HCC cells to sorafenib [131]. ADRB2 signaling promotes sorafenib resistance by autophagic degradation of HIF1α, and ADRB2 inhibitor ICI118551 could overcome sorafenib resistance in HCC cells [106]. However, it was demonstrated that sorafenib treatment can lead to a switch from the HIF1α- to HIF2α-dependent pathway, requiring inhibition of both to overcome sorafenib resistance, which could be achieved by treatment with 2-Methoxyestradiol (2ME2) [157]. It was shown that the sorafenib-induced HIF1α- to HIF2α switch resulted in activation of the TGFα/EGFR pathway, and the EGFR inhibitor gefitinib could synergize with sorafenib to induce apoptosis in hypoxic HCC cells [158]. An HIF2α inhibitor, PT-2385, significantly augmented sorafenib efficacy in the orthotopic mouse model of HCC [159].

### 5.3. Epigenetic Modulation

SHELTER is a phase I/II trial in 57 HCC patients, with radiologically confirmed progression on sorafenib, to evaluate the efficacy of sorafenib and the HDAC inhibitor resminostat combination [160]. PFS and OS for the combination was 62.5% and 8 months versus 12.5% and 4.1 months for resminostat alone, with manageable safety, suggesting that this combination might be evaluated in a larger trial [160]. KDM1A inhibitor GSK2879552 was shown to revert CSC properties and resensitized HCC cells to sorafenib in vivo [82].

### 5.4. Development of Novel Next-Generation TKIs

Newer generations of TKIs are being developed with enhanced specificity, potency, and potentially less severe serious adverse events (SAEs). The FGF19/FGFR4 pathway has been identified as an HCC driver [161]. Fisogatinib/BLU-554 is a selective oral FGFR4 inhibitor which was evaluated in a phase I dose escalation study in advanced HCC patients using FGF19 as a biomarker in immunohistochemical staining of the tumor [162]. In 81 patients, fisogatinib was well tolerated with manageable adverse events, and the overall response rate was 17% in FGF19-positive patients compared to 0% in FGF19-negative patients [162]. A phase Ib/II trial using a combination of BLU-554 and anti-PD-L1 antibody (CS1001) was carried out in 18 advanced or metastatic HCC patients, including 8 FGF19-positive patients [163]. The patients showed grade I and II adverse events with 100% disease control rate (DCR) and 50% objective response rate (ORR), with FGF19-positive patients showing the best response [163]. These two studies are promising for ~15% HCC patients who harbor genomic amplification of FGF19 gene in chromosome 11q13.3, and mandates evaluation of fisogatinib in phase III trials in a larger cohort of patients [161]. Apatinib/rivoceranib is a predominantly VEGFR-2-specific inhibitor which was evaluated in combination with anti-PD-1 antibody camrelizumab against sorafenib in a phase III randomized, open label trial (CARES-310), in patients with unresectable HCC, with 272 receiving the combination while 271 receiving sorafenib [164]. The combination showed significant and clinically meaningful benefit in progression-free survival (PFS; 5.6 months vs. 3.7 months) and median overall survival (22.1 months vs. 15.2 months) compared to sorafenib alone [164]. However, treatment-related SAEs were reported in 24% of the patients in the combination group vs. 6% of the patients in the sorafenib-alone group [164]. Nevertheless, based on this study, apatinib+camrelizumab combination has been approved as the first line of treatment for inoperable HCC in China. Additional combination approaches that include apatinib are being evaluated in advanced HCC, demonstrating promising survival benefits [165,166]. Anlotinib hydrochloride (AHC) is an oral TKI targeting VEGFR1-3, FGFR1-4, and PDGFRα/β [167]. In a phase II trial in 38 intermediate or advanced HCC, the efficacy of transcatheter arterial chemoembolization (TACE) alone or TACE+AHC was evaluated [168]. Median progression-free survival (PFS) was significantly higher in the TACE+AHC group compared to TACE alone (11.04 months vs. 6.87 months, respectively), with patients maintaining quality of life [168]. A number of MET inhibitors, such as tivantinib, tepotinib, and capmatinib, have been evaluated in advanced HCC patients with especially high MET expression and previously treated with sorafenib [169,170,171]. The result of these clinical trials in prolonging OS, however, was not significant; the manageable safety profiles suggest that further interrogation of these inhibitors is worth pursuing.

### 5.5. Gene Therapy Approaches

Gene therapy approaches using oncolytic viruses are increasingly being recognized as an alternative treatment for cancer [172]. Oncolytic viruses are genetically altered viruses that specifically target tumor cells for infection with subsequent lysis of tumor cells with little to no damage to healthy tissues [173]. Pexa-vec (pexastimogene devacirepvec, JX-594) is an oncolytic vaccinia virus which selectively infects and replicates in cancer cells, causing lysis of cancer cells and also expresses human granulocyte–macrophage colony-stimulating factor (hGM-CSF), which stimulates a long-term anti-tumor immunity [174]. In a randomized phase IIa trial in predominantly sorafenib-naïve advanced HCC patients, pexa-vec demonstrated dose-related survival advantage, with the median survival of the patients being 14.1 months and 6.7 months with high and low doses, respectively [175]. In a randomized, open-label phase IIb study (TRAVERSE) in 129 HCC patients who failed sorafenib treatment, pexa-vec was tested with best supportive care (BSC) [176]. Despite a tolerable safety profile and induction of anti-tumor T cell response, pexa-vec did not improve overall survival [176]. PHOCUS is a phase III, randomized open-label study at 142 sites in 16 countries that compared sequential pexa-vec and sorafenib treatment with sorafenib alone in 459 advanced HCC patients with no prior systemic treatment [177]. However, the combination did not show improved clinical benefit over sorafenib and in some patients the combination exerted more serious adverse effect over sorafenib, resulting in early termination of the study [177]. Although these clinical trials did not indicate any clear benefit with pexa-vec, they provide fundamental information about selection of patients and patients’ clinical criteria that might help design better trials in future. The TRAVERSE trial was performed in patients who failed sorafenib treatment, i.e., these patients had primary sorafenib resistance [176]. It may be valuable to carry out a clinical trial with pexa-vec in HCC patients who developed acquired sorafenib resistance. GLV-1h68 is an oncolytic vaccinia virus which did not express GM-CSF, rather had expression cassettes encoding *Renilla* luciferase–*Aequorea* green fluorescent protein fusion, β-galactosidase, and β-glucuronidase, and showed efficient killing of sorafenib-resistant SNU-739, SNU-449, HuH-7, and Hep3B cells in vitro [178]. It would be interesting to see whether these in vitro findings are translatable in pre-clinical mouse studies as well as in clinical trials.

## 6. Conclusions and Future Perspectives

### 6.1. Summary of Key Findings

Overcoming resistance to TKIs in HCC remains a significant challenge in oncology. It is important to understand the molecular factors associated with the development of resistance to TKIs and this will help us develop new therapeutic strategies based on the molecular target and reduce the relapse rate. Resistance mechanisms, including genetic mutations, activation of alternative signaling pathways, and changes in the TME, complicate treatment. However, advances in the understanding of these mechanisms have led to the development of new strategies aimed at mitigating resistance. The multi-faceted approach includes the development of next-generation TKIs with enhanced specificity and potency; combination therapies that leverage the synergistic effects of TKIs with other treatments such as ICIs and targeted agents; modulation of the TME to disrupt the supportive environment for tumor cells; and implementation of alternative strategies such as gene-based therapies. Personalized medicine, guided by biomarker identification and genomic profiling, offers tailored treatment strategies to improve outcomes. Enhanced drug delivery systems and continuous monitoring of drug levels and tumor response can further optimize therapy.

### 6.2. Future Directions and Research Priorities

One major challenge of successful application of the aforementioned strategy is employment of appropriate models in pre-clinical studies so that the findings can be faithfully translated in clinical trials. HCC risk factors include viral hepatitis, alcoholism, and MASH, all of which lead to chronic inflammation [179]. Studying HCC in an etiology-driven mouse model is difficult, as HBV- and HCV-transgenic mice either do not develop HCC or develop at a very late age with low and variable frequency [180,181,182,183,184]. Diethylnitrosamine (DEN)-induced HCC is an ideal surrogate model because the hepatocyte injury caused by DEN and subsequent inflammatory changes mimic the initial events in human HCC [185,186,187,188,189,190,191,192]. HCC usually develops on a cirrhotic liver with extensive fibrosis. CCl_4_ injection subsequent to DEN is an established model of fibrotic HCC [193,194,195]. The National Institute for Occupational Safety and Health (NIOSH) estimates that 58,208 workers are potentially exposed to CCl_4_ in the US, thus establishing the relevance of this model to human disease [196]. Long-term feeding with a high fat/high sugar (HF/HS) diet results in HCC development, serving as a model for obesity-induced HCC [197]. Although these models are long, they recapitulate human HCC development and progression, and designing an appropriate experimental scheme has the likelihood of generating robust meaningful findings to have an impact on the management of human HCC. One potential alternative to these long approaches could be treating the mice with CCl_4_ or feeding them an HF/HS diet to create an inflammatory/fibrotic environment, and then establishing an orthotopic model of HCC using established HCC cell lines or HCC PDXs, or initiating oncogene-induced HCC by hydrodynamic injection [198,199,200].

The development of next-generation TKIs with broader target spectra and improved efficacy against resistant cancer cell populations will be crucial. These agents should be designed to target multiple pathways simultaneously to reduce the likelihood of resistance through alternative signaling. Future research should focus on identifying the most effective combinations of TKIs with other therapeutic agents. Clinical trials exploring various combinations will be essential in determining optimal treatment protocols and uncovering new synergistic effects. Developing therapies that specifically target components of the TME, such as cancer-associated fibroblasts (CAFs), immune cells, and the extracellular matrix (ECM), will be vital. Agents that can modulate the TME to reduce tumor support and enhance TKI efficacy should be prioritized. Exploring epigenetic therapies, such as histone deacetylase (HDAC) inhibitors and DNA methyltransferase (DNMT) inhibitors, in combination with TKIs can provide new avenues to overcome resistance. Research in this area should aim to identify the most effective epigenetic targets. Advances in genomic profiling and biomarker discovery will enhance the ability to tailor treatments to individual patients (personalized medicine). Future efforts should focus on integrating these technologies into clinical practice to enable real-time, adaptive treatment strategies based on patient-specific tumor characteristics. The development of innovative drug delivery systems, such as nanoparticle-based carriers and conjugated antibody-TKI complexes, will improve drug bioavailability and target specificity. Research should continue to optimize these technologies for clinical use. Ongoing clinical trials and translational research are critical to advancing the understanding of TKI resistance mechanisms and identifying new therapeutic targets.

## Figures and Tables

**Figure 1 cancers-16-03944-f001:**
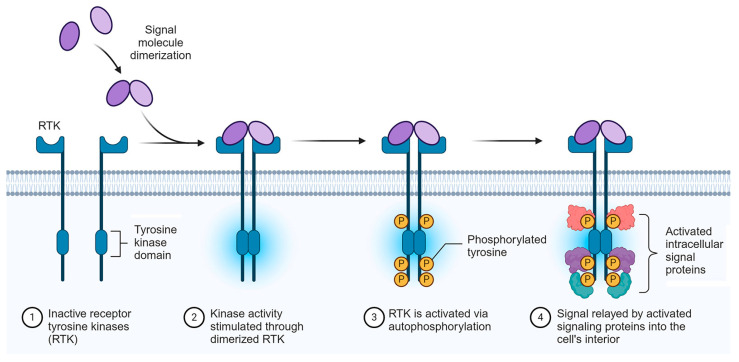
Mechanism of receptor tyrosine kinase (RTK) activation. Created in BioRender.com.

**Figure 2 cancers-16-03944-f002:**
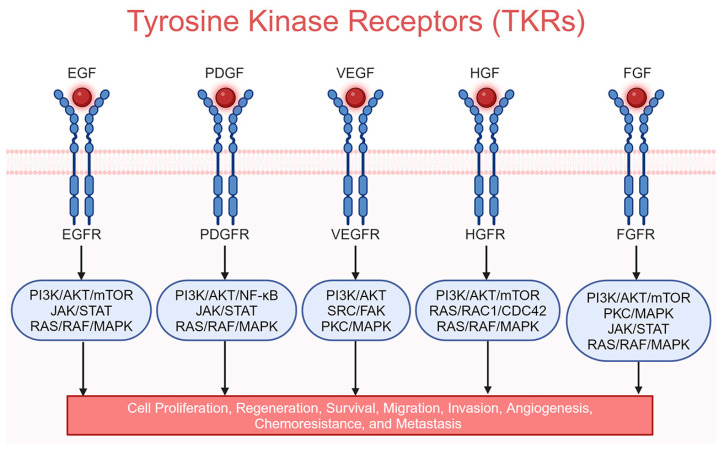
Cartoon showing common tyrosine kinase receptors and their downstream signaling pathways. Created in BioRender.com.

**Figure 3 cancers-16-03944-f003:**
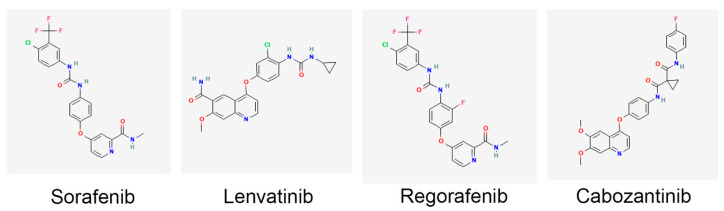
Chemical structures of commonly used TKIs approved by the FDA for treatment of HCC. Created in BioRender.com.

**Figure 4 cancers-16-03944-f004:**
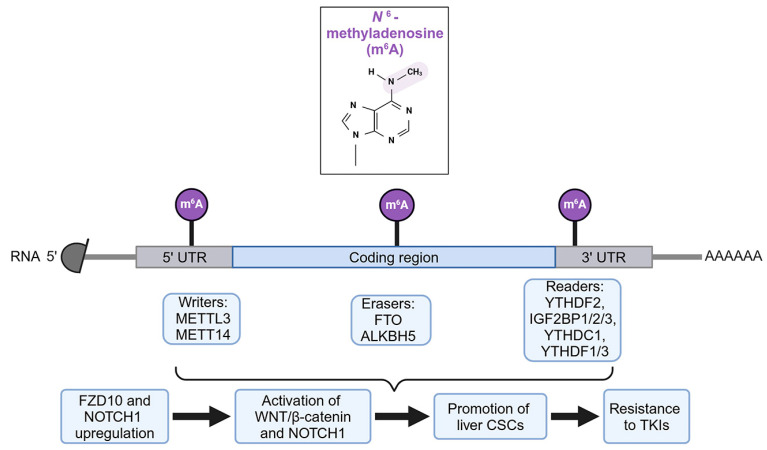
Schematic of m6A RNA modification and its contribution to resistance to TKIs. See text for details. Created in BioRender.com.

**Figure 5 cancers-16-03944-f005:**
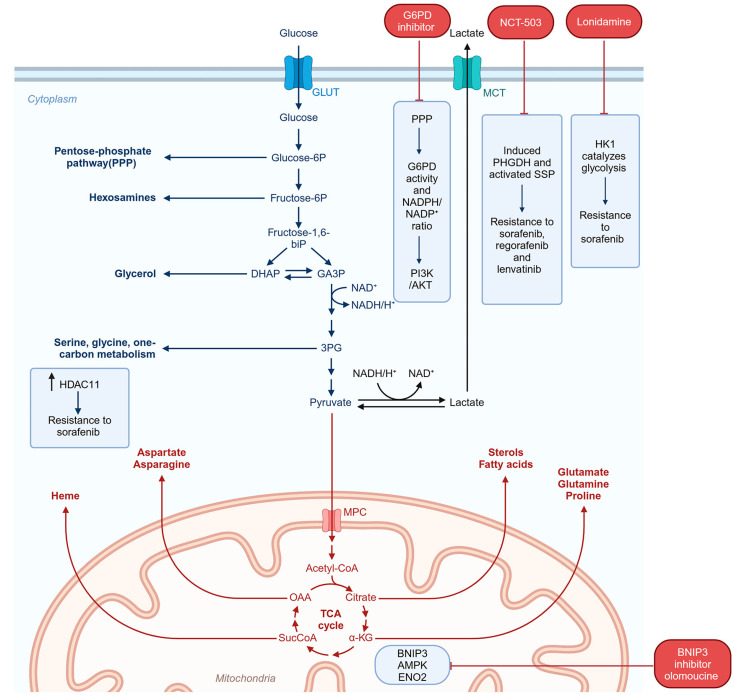
Schematic of metabolic changes contributing to resistance to TKIs. See text for details. Created in BioRender.com.

**Figure 6 cancers-16-03944-f006:**
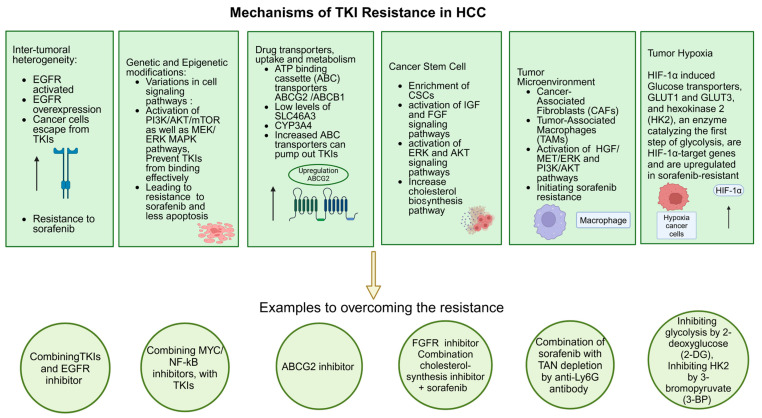
Important mechanisms of TKI resistance and strategies to overcome them. Please see text for details. Created in BioRender.com.

**Table 1 cancers-16-03944-t001:** miRNAs that mediate sorafenib resistance in HCC.

miRNAs	Expression	Target	Mechanism	References
miR-622	Decreased	KRAS	Activation of PI3K/AKT and MEK/ERK pathways	[114]
miR-181a	Increased	RASSF1	Activation of MAPK pathway	[115]
miR-1991-5p and let-7C	Decreased	MAPK4K3	Activation of MAPK pathway	[116]
miR-21, miR-222, miR-494	Increased	PTEN	Activation of PI3K/AKT pathway	[117,119,120]
miR-216a/217	Increased	SMAD7 and PTEN	Activation of TGFβ and PI3K/AKT pathways	[118]
miR-93	Increased	PTEN and CDKN1A	Activation of MET and PI3K/AKT pathways	[121]
miR-7	Decreased	TYRO3	Activation of PI3K/AKT pathway	[122]
miR-221	Increased	Caspase-3	Inhibition of apoptosis	[123]
miR-193b	Decreased	Mcl-1	Inhibition of apoptosis	[124]
miR-34a	Decreased	Bcl-2	Inhibition of apoptosis	[125]
let-7 family	Decreased	Bcl-xL	Inhibition of apoptosis	[126]
miR-142-3p	Decreased	ATG5 and ATG16L1	Increased autophagy	[127]
miR-122	Decreased	IGF-1R and SLC7A1	Activation of IGF signaling and increased nitric oxide	[128,129]

**Table 2 cancers-16-03944-t002:** Upregulated lncRNAs and circRNAs mediating TKI-resistance in HCC.

lncRNAs	Target	Mechanism	References
MALAT1	miR-140-5p	Increased Aurora kinase A	[134]
THOR	β-catenin	Stabilizes β-catenin and increases CSCs	[135]
NEAT1	miR-335	Activates c-MET and AKT pathways	[136]
TRERNA1	miR-22-3p	Increased NRAS and activation of RAF/MEK/ERK pathway	[137]
LINC-ROR	miR-876-5p	Increased FOXM1	[138]
ZFAS1	Not identified	Increased stemness and EMT	[139]
linc-VLDLR	Not identified	Increased ABCG2	[140]
BBOX1-AS1	miR-361-3p	Increased PHF8	[141]
URB1-AS1	Ferritin	Inhibition of ferroptosis	[142]
circRNA-SORE	miR-103a-2-5p and miR-660-3p; YBX1	Activation of Wnt/β-catenin signaling pathway; stabilization of YBX1	[143,144]
circFOXM1	miR-1324	Increased MECP2	[145]
circDCAF8	miR-217	Increased NAP1L1	[146]

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
