# Peer review of "Resistance to Tyrosine Kinase Inhibitors in Hepatocellular Carcinoma (HCC): Clinical Implications and Potential Strategies to Overcome the Resistance"

_cancers, 2024, doi:10.3390/cancers16233944_

Round 1
Reviewer 1 Report
Comments and Suggestions for Authors
Ermi and Sarkar review the universal and inevitable development of resistance to TKIs in HCC. This is a very good review of the field. However, the IMbrave150 clinical trial demonstrated the superiority of anti-PD-L1 antibody atezolizumab and the anti-VEGF antibody bevacizumab combination over sorafenib in terms of OS and PFS in treatment naïve patients. This trial has changed the therapeutic approach in liver cancer. This review should address upfront why it is important to investigate resistance to TKI in liver cancer when they are no longer the first choice of treatment. For example, are there patients non eligible for the atezolizumab and bevacizumab combination that still require TKI. In addition, the following points are suggested for improvement.
1- The mechanisms explained include the activation of EGFR and the RAS/RAF/MEK/ERK pro-survival signaling pathways, that also includes NF1 and DUSP9 and the PI3K/mTOR pathway. The authors cite studies that in general support combination therapies between TKIs and inhibitors of these pathways. However, given the availability of these drugs for clinical trials, it is surprising that trials testing those combinations were not considered. A phase trial conducted by Kim et al PMID: 32776632, concluded that the combination has limited anticancer activity. Another trial used selumetinib, a competitive MEK1/2 inhibitor, (NCT00604721) including patients with locally advanced HCC but it was halted at the first interim analysis as no patients demonstrated a radiographic tumor response. This is not consistent with the narrative supported in the review that resistance is mostly dependent of EGFR and RAS-MAPK pathways. Obviously additional factors causing resistant to sorafenib must be discussed including recent non genetic mechanisms where ‘persisters’, cells with non-genetic adaptations to drug actions contribute to tumor relapse after treatment.
2- Fig 1 and 3 present notions that has been illustrated and reviewed multiple times and is biased to the classic pathwas &ERK, PI3K) that has failed in clinical trials. However, new highly interesting concepts presented in the review such as the role of m6A mRNA modification and metabolism in drug resistance, may benefit from summarizing schematics.
3- In page 10: Interestingly, serine per se could not confer TKI resistnace to HCC cells, suggesting that additional metabolites generated through SSP might mediate the resistance [87].
4- A table summarizing all lncRNAs and cirRNAs linked to TKI resistance in HCC would be helpful.
5- Consider moving the hypoxia section to the metabolism section.
Author Response
We thank the reviewers for their astute comments and suggestions and finding our manuscript as a very good review of the field. We addressed the reviewers’ comments and concerns editing the text and adding two new figures, Figures 4 and 5, and two tables, Tables 1 and 2, in the revised manuscript. We believe the revised manuscript is significantly improved. A point-by-point response is provided below:
Reviewer#1
- Ermi and Sarkar review the universal and inevitable development of resistance to TKIs in HCC. This is a very good review of the field. However, the IMbrave150 clinical trial demonstrated the superiority of anti-PD-L1 antibody atezolizumab and the anti-VEGF antibody bevacizumab combination over sorafenib in terms of OS and PFS in treatment naïve patients. This trial has changed the therapeutic approach in liver cancer. This review should address upfront why it is important to investigate resistance to TKI in liver cancer when they are no longer the first choice of treatment. For example, are there patients non-eligible for the atezolizumab and bevacizumab combination that still require TKI.
Answer: We thank the reviewer for raising this important point. We have now addressed this issue in section 1.2, lines 87-99 in the revised manuscript. We have also added sentences in the Abstract to describe the importance of TKIs, even though immunotherapy is currently recommended as the first line of treatment.
- The mechanisms explained include the activation of EGFR and the RAS/RAF/MEK/ERK pro-survival signaling pathways, that also includes NF1 and DUSP9 and the PI3K/mTOR pathway. The authors cite studies that in general support combination therapies between TKIs and inhibitors of these pathways. However, given the availability of these drugs for clinical trials, it is surprising that trials testing those combinations were not considered. A phase trial conducted by Kim et al PMID: 32776632, concluded that the combination has limited anticancer activity. Another trial used selumetinib, a competitive MEK1/2 inhibitor, (NCT00604721) including patients with locally advanced HCC but it was halted at the first interim analysis as no patients demonstrated a radiographic tumor response. This is not consistent with the narrative supported in the review that resistance is mostly dependent of EGFR and RAS-MAPK pathways. Obviously additional factors causing resistant to sorafenib must be discussed including recent non-genetic mechanisms where ‘persisters’, cells with non-genetic adaptations to drug actions contribute to tumor relapse after treatment.
Answer: We completely agree with the reviewer. There are many in vitro studies showing potential role of EGFR and RAS/RAF/MEK/ERK pathway activation contributing to TKI resistance. In Section 3.2 we discussed these in vitro studies and then described the SEARCH trial demonstrating no effect of erlotinib+sorafenib combination in 720 patients. We have now cited the Kim et al. paper, as suggested by the reviewer, which evaluated sorafenib+trametinib combination in 17 treatment naïve HCC patients without showing any response. However, it seems like for lenvatinib the scenario is somewhat different where it was shown that for both preclinical studies using HCC PDXs and a small clinical trial using 12 patients, Lenvatinib+gefitinib combination exhibited meaningful clinical response. Similarly, trametinib could overcome Lenvatinib resistance induced by NF1 and DUSP9 (described in second paragraph of Section 3.3). In Section 3.2 we discussed these differences between sorafenib and lenvatinib in regards to EGFR inhibitors. The ‘persisters’ have chromatin-mediated epigenetic changes which are described in Section 3.4. We introduce the concept of drug-tolerant expanded persisters (DTEPs) in this section. However, we did not expand on this concept more because we mentioned at the end of the first paragraph of Section 3, ‘In this review we discuss TKI resistance mechanisms that have been identified in HCC without extrapolating observations from other cancers or other drugs.’ DTEPs have been confirmed in multiple cancer types, including breast, prostate, gastric and colon cancers and melanoma, but not yet in HCC (PMID: 38508142).
- Fig 1 and 3 present notions that has been illustrated and reviewed multiple times and is biased to the classic pathways &ERK, PI3K) that has failed in clinical trials. However, new highly interesting concepts presented in the review such as the role of m6A mRNA modification and metabolism in drug resistance, may benefit from summarizing schematics.
Answer: We thank the reviewer for the suggestion and now have created two new figures, Figures 4 and 5, in the revised manuscript showing these schematics.
- In page 10: Interestingly, serine per se could not confer TKI resistance to HCC cells, suggesting that additional metabolites generated through SSP might mediate the resistance [87].
Answer: We apologize for the typo which has been corrected in the revised manuscript.
- A table summarizing all lncRNAs and cirRNAs linked to TKI resistance in HCC would be helpful.
Answer: We thank the reviewer for the suggestion and new tables, Tables 1 and 2, are included in the revised manuscript.
- Consider moving the hypoxia section to the metabolism section.
Answer: We have moved the hypoxia section to the metabolism section as suggested by the reviewer.
Reviewer 2 Report
Comments and Suggestions for Authors
1- There is a fundamental problem in the first part of the paper, in particular the Abstract and parts of the Introduction (lines 99-118). These passages emphasize the role of angiogenesis and give the impression that TKIs mainly, if not exclusively, act on angiogenesis. This is clearly not so.
Only lines 184 / 197 finally introduce the tumor cell as major driver of oncogenesis, resistance and site of TKI action. Thus, the said introductory parts are misleading and need substantial re-writing to re-focus those from angiogenesis towards the HCC cell.
2- It seems that Abstract and Simple Summary are flipped
3- Different parts of the manuscript were clearly written by different authors. In some parts (but not in others), many commas are missing. In some parts (but not in others), many articles (‘the’) are missing. Examples: Among these mRNAs*,* CDK1…; revealed that *the* pentose phosphate pathway, etc. Check throughout the manuscript for many missing ‘the’, missing commas etc.
4- Section 3.5.: It is unclear from the description of [64] how plasma levels should connect to efflux out of HCC cells
5- Figure 4: the text in the figure is too small. Increase the font size
6- Multiple minor issues require correction:
Line 216: remain*s* to be seen
Line 317: complex
Line 407: mon*o*
Line 464: xenograft*s*
Line 518: sorafenib *treated* HCC patients
Lines 570-572: the contents of this sentence are unclear
Line 626: con tribute*e*d
Line 661: tumorigenic M1 type *or* anti-inflammatory
Line 713: subcutaneous*
Line 727: in-roads.*
Line 740: combination
Line 757-761: poor sentence structure, please rephrase
Line 833: . *Although
Line 860: resistance
Line 860: might be worthier carrying out – incorrect grammar, please rephrase
Author Response
Reviewer#2
- There is a fundamental problem in the first part of the paper, in particular the Abstract and parts of the Introduction (lines 99-118). These passages emphasize the role of angiogenesis and give the impression that TKIs mainly, if not exclusively, act on angiogenesis. This is clearly not so. Only lines 184 / 197 finally introduce the tumor cell as major driver of oncogenesis, resistance and site of TKI action. Thus, the said introductory parts are misleading and need substantial re-writing to re-focus those from angiogenesis towards the HCC cell.
Answer: We thank the reviewer for raising this important comment. We’ve modified the text in second paragraph of Section 2 as well as in Simple Summary to address this issue.
- It seems that Abstract and Simple Summary are flipped
Answer: Actually, it was not. The Simple Summary was longer because we needed to explain in lay language. In the revised manuscript the abstract has been expanded.
- Different parts of the manuscript were clearly written by different authors. In some parts (but not in others), many commas are missing. In some parts (but not in others), many articles (‘the’) are missing. Examples: Among these mRNAs*,* CDK1…; revealed that *the* pentose phosphate pathway, etc. Check throughout the manuscript for many missing ‘the’, missing commas etc.
Answer: We apologize for these errors. The revised manuscript has been thoroughly proof-read.
- Section 3.5.: It is unclear from the description of [64] how plasma levels should connect to efflux out of HCC cells.
Answer: We agree with the reviewer’s comment and removed the connection with drug efflux in the revised manuscript.
- Figure 4: the text in the figure is too small. Increase the font size
Answer: We have enlarged Figure 4 so the text is legible.
- Multiple minor issues require correction:
Line 216: remain*s* to be seen
Line 317: complex
Line 407: mon*o*
Line 464: xenograft*s*
Line 518: sorafenib *treated* HCC patients
Lines 570-572: the contents of this sentence are unclear
Line 626: con tribute*e*d
Line 661: tumorigenic M1 type *or* anti-inflammatory
Line 713: subcutaneous*
Line 727: in-roads.*
Line 740: combination
Line 757-761: poor sentence structure, please rephrase
Line 833: . *Although
Line 860: resistance
Line 860: might be worthier carrying out – incorrect grammar, please rephrase
Answer: We thank the reviewer for the meticulous reading and highlighting our errors. The manuscript has been thoroughly checked for typos and other minor errors which have been corrected.